# Deep Generative Classifier for Out-of-Distribution Sample Detection

## Abstract

The capability of reliably detecting out-of-distribution samples is one of the key factors in deploying a good classifier, as the test distribution always does not match with the training distribution in most real-world applications. In this work, we propose a deep generative classifier which is effective to detect out-of-distribution samples as well as classify in-distribution samples, by integrating the concept of Gaussian discriminant analysis into deep neural networks. Unlike the discriminative (or softmax) classifier that only focuses on the decision boundary partitioning its latent space into multiple regions, our generative classifier aims to explicitly model class-conditional distributions as separable Gaussian distributions. Thereby, we can define the confidence score by the distance between a test sample and the center of each distribution. Our empirical evaluation on multi-class images and tabular data demonstrate that the generative classifier achieves the best performances in distinguishing out-of-distribution samples, and also it can be generalized well for various types of deep neural networks.

## 1 Introduction

Out-of-distribution (OOD) detection, also known as novelty detection, refers to the task of identifying the samples that differ in some respect from the training samples. Recently, deep neural networks (DNNs) turned out to show unpredictable behaviors in case of mismatch between the training and testing data distributions; for example, they tend to make high confidence prediction for the samples that are drawn from OOD or belong to unseen classes (Szegedy et al., 2014; Moosavi-Dezfooli et al., 2017). For this reason, accurately measuring the *distributional uncertainty* (Malinin & Gales, 2018) of DNNs becomes one of the important challenges in many real-world applications where we can hardly control the testing data distribution. Several recent studies have tried to simply detect OOD samples using the confidence score defined by softmax probability (Hendrycks & Gimpel, 2017; Liang et al., 2018) or Mahalanobis distance from class means (Lee et al., 2018), and they showed promising results even without re-training the model.

However, all of them employ the DNNs designed for a discriminative (or softmax) classifier, which has limited power to locate OOD samples distinguishable with in-distribution (ID) samples in their latent space. To be specific, the softmax classifier is optimized to learn the discriminative latent space where the training samples are aligned along their corresponding class weight vectors, maximizing the softmax probability for the target classes. As pointed out in (Hendrycks & Gimpel, 2017), OOD samples are more likely to have small values of the softmax probability for all known classes, which means that their latent vectors get closer to the origin. As a result, there could be a large overlap between two sets of ID and OOD samples in the latent space (Figure 1), which eventually reduces the gap between their confidence scores and degrades the performance as well.

In addition, most of existing confidence scores adopt additional calibration techniques (Goodfellow et al., 2014; Hinton et al., 2015) to enhance the reliability of the detection, but they include several hyperparameters whose optimal values vary depending on the testing data distribution. In this situation, they utilized a small portion of each test set (containing both ID and OOD samples) for validation, and reported the results evaluated on the rest by using the optimal hyperparameter values for each test case. Considering the motivation of OOD detection that prior knowledge of test distributions is not available before we encounter them, such process of tuning the hyperparameters for each test case is not practical when deploying the DNNs in practice.

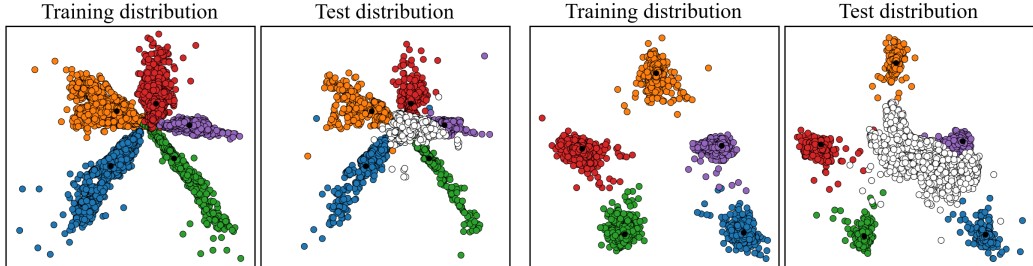

Figure 1: The 2D latent spaces obtained by the softmax classifier (Left) and our proposed generative calssifier (Right). In-distribution and out-of-distribution samples are shown as colored and white circles, respectively. (Dataset: GasSensor, OOD class: "Ammonia", Model: Multi-layer perceptron)

In this paper, we propose a novel objective to train DNNs with a generative (or distance) classifier which is capable of effectively identifying OOD test samples. The main difference of our deep generative classifier is to learn separable class-conditional distributions in the latent space, by explicitly modeling them as a DNN layer. The generative classifier places OOD samples further apart from the distributions of all given classes, without utilizing OOD samples for its validation. Thus, based on the Euclidean distance between a test sample and the centers of the obtained class-conditional distributions, we can calculate how likely and how confidently the sample belongs to each class. This can be interpreted as a multi-class extension of unsupervised anomaly detection (Ruff et al., 2018), and Gaussian discriminant analysis provides the theoretical background for incorporating the generative classifier into the DNNs. Our extensive experiments on images and tabular data demonstrate that the proposed classifier distinguishes OOD samples more accurately than the state-of-the-art method, while maintaining the classification accuracy for ID samples.

## 2 METRIC LEARNING-BASED DEEP GENERATIVE CLASSIFIER

We introduce a novel objective for training deep neural networks (DNNs) with a generative classifier, which is able to effectively detect out-of-distribution samples as well as classify in-distribution samples into known classes. We first derive the learning objective from the Gaussian discriminant analysis, and propose the distance-based confidence score for out-of-distribution sample detection.

**Metric learning objective for classification.** The key idea of our objective is to optimize the deep learning model so that the latent representations (i.e., the outputs of the last layer) of data samples in the same class gather together thereby form an independent sphere. In other words, it aims to learn each class-conditional distribution in the latent space to follow a normal distribution that is entirely separable from other class-conditional distributions. Using the obtained distributions, we can calculate the class-conditional probabilities that indicate how likely an input sample is generated from each distribution, and this probability can serve as a good measure of the confidence. We define the two terms based on the Euclidean distance between the data representations obtained by the DNNs, denoted by $f(\mathbf{x})$, and the center of each class-conditional distribution, denoted by $\mathbf{c}_k$. Given $N$ training samples $\{(\mathbf{x}_1, y_1), \ldots, (\mathbf{x}_N, y_N)\}$ from $K$ different classes, the objective is described as follows.

$$\min_{\mathcal{W}, \mathbf{c}, b} \frac{1}{N} \sum_{i=1}^{N} \left( -\log \frac{\exp(-\|f(\mathbf{x}_i; \mathcal{W}) - \mathbf{c}_{y_i}\|^2 + b_{y_i})}{\sum_{k=1}^{K} \exp(-\|f(\mathbf{x}_i; \mathcal{W}) - \mathbf{c}_k\|^2 + b_k)} + \lambda \cdot \|f(\mathbf{x}_i; \mathcal{W}) - \mathbf{c}_{y_i}\|^2 \right), \quad (1)$$

The objective includes three types of trainable parameters: the weights of DNNs $\mathcal{W}$, the class centers $\mathbf{c}_1, \ldots, \mathbf{c}_K$ and biases $b_1, \ldots, b_K$. All of them can be effectively optimized by stochastic gradient descent (SGD) and back-propagation, which are widely used in deep learning.

Note that we directly optimize the latent space induced by the DNNs using Euclidean distance, similarly to other metric learning objectives. Existing deep metric learning based on the triplet loss (Hoffer & Ailon, 2015; Schroff et al., 2015) learns the distance among training samples utilizing their label information to capture their similarities into the metric space for a variety of retrieval

tasks. On the other hand, our objective focuses on the distance between the samples and their target class centers for the accurate modeling of class-conditional distributions.

**Derivation from Gaussian discriminant analysis.** Our objective for the generative classifier can be understood from the perspective of Gaussian discriminant analysis (GDA) (Murphy, 2012). The generative classifier defines the posterior distribution $P(y|\mathbf{x})$ by using the class-conditional distribution $P(\mathbf{x}|y)$ and class prior $P(y)$. In case of GDA, each class-conditional distribution is assumed to follow the multivariate Gaussian distribution (i.e., $P(\mathbf{x}|y = k) = \mathcal{N}(\mathbf{x}|\mu_k, \Sigma_k)$) and the class prior is assumed to follow the Bernoulli distribution (i.e., $P(y = k) = \frac{\beta_k}{\sum_{k'} \beta_{k'}}$). To simply fuse GDA with DNNs, we further fix all the class covariance matrices to the identity matrix (i.e., $\Sigma_k = I$). Then, the posterior probability that a sample $f(\mathbf{x})$ belongs to the class $k$ is described as

$$P(y = k|f(\mathbf{x})) = \frac{P(y = k)P(f(\mathbf{x})|y = k)}{\sum_{k'} P(y = k')P(f(\mathbf{x})|y = k')} = \frac{\exp(-\frac{1}{2}\|f(\mathbf{x}) - \mu_k\|^2 + \log \beta_k)}{\sum_{k'} \exp(-\frac{1}{2}\|f(\mathbf{x}) - \mu_{k'}\|^2 + \log \beta_{k'})}.$$

Considering $\mu_k$ and $\log \beta_k$ as the class center $\mathbf{c}_k$ and bias $b_k$ respectively, the first term of our objective (1) is equivalent to the negative log posterior probability. That is, the objective eventually trains the classifier by maximizing the posterior probability for training samples.

However, the direct optimization of the DNNs and other parameters by its gradient does not guarantee that the class-conditional distributions become the Gaussian distributions and the class centers are the actual class means of training samples. Thus, to enforce our GDA assumption, we minimize the Kullback-Leibler (KL) divergence between the $k$-th empirical class-conditional distribution and the Gaussian distribution whose mean and covariance are $\mathbf{c}_k$ and $I$, respectively. The empirical class-conditional distribution is represented by the average of the dirac delta functions for all training samples of a target class, i.e., $\mathbb{P}_k = \frac{1}{N_k} \sum_{y_i = k} \delta(\mathbf{x} - f(\mathbf{x}_i))$, where $N_k$ is the number of the training samples of the class $k$. Then, the KL divergence is formulated as

$$\mathrm{KL}(\mathbb{P}_k \parallel \mathcal{N}(\mathbf{c}_k, I)) = -\int \frac{1}{N_k} \sum_{y_i = k} \delta(\mathbf{x} - f(\mathbf{x}_i)) \log\left[\frac{1}{(2\pi)^{d/2}} \exp\left(-\frac{1}{2}\|\mathbf{x} - \mathbf{c}_k\|^2\right)\right] \mathrm{d}\mathbf{x}$$

$$+ \int \frac{1}{N_k} \sum_{y_i = k} \delta(\mathbf{x} - f(\mathbf{x}_i)) \log\left[\frac{1}{N_k} \sum_{y_i = k} \delta(\mathbf{x} - f(\mathbf{x}_i))\right] \mathrm{d}\mathbf{x}$$

$$= -\frac{1}{N_k} \sum_{y_i = k} \log\left[\frac{1}{(2\pi)^{d/2}} \exp\left(-\frac{1}{2}\|f(\mathbf{x}_i) - \mathbf{c}_k\|^2\right)\right] + \log \frac{1}{N_k}$$

$$= \frac{1}{2N_k} \sum_{y_i = k} \|f(\mathbf{x}_i) - \mathbf{c}_k\|^2 + \text{constant}.$$

The entropy term of the empirical class-conditional distribution can be calculated by using the definition of the dirac measure (Murphy, 2012). By minimizing this KL divergence for all the classes, we can approximate the $K$ class-conditional Gaussian distributions. Finally, we complete our objective by combining this KL term with the posterior term using the $\lambda$-weighted sum in order to control the effect of the regularization. We remark that $\lambda$ is the hyperparameter used for training the model, which depends on only ID, not OOD; thus it does not need to be tuned for different test distributions.

**In-distribution classification.** Since our objective maximizes the posterior probability for the target class of each sample $P(y = y_i|\mathbf{x})$, we can predict the class label of an input sample to the class that has the highest posterior probability as follows.

$$\hat{y}(\mathbf{x}) = \arg\max_k P(y = k|\mathbf{x}) = \arg\max_k \left(-\|f(\mathbf{x}) - \mathbf{c}_k\|^2 + b_k\right) \tag{2}$$

In terms of DNNs, our proposed classifier replaces the fully-connected layer (fc-layer) computing the final classification score by $\mathbf{w}_k \cdot f(\mathbf{x}) + b_k$ with the *distance metric layer* (dm-layer) computing the distance from each center by $-\|f(\mathbf{x}) - \mathbf{c}_k\|^2 + b_k$. In other words, the class label is mainly predicted by the distance from each class center, so we use the terms "distance classifier" and "generative classifier" interchangeably in the rest of this paper. The dm-layer contains the exactly same number of model parameters with the fully-connected layer, because only the weight matrix $W = [\mathbf{w}_1; \ldots; \mathbf{w}_K] \in \mathbb{R}^{K \times d}$ is replaced with the class center matrix $C = [\mathbf{c}_1; \ldots; \mathbf{c}_K] \in \mathbb{R}^{K \times d}$.

Table 1: Statistics of tabular datasets.

| Dataset | # Attributes | # Instances | # Classes |
|---|---|---|---|
| GasSensor | 128 | 13,910 | 6 |
| Shuttle | 9 | 58,000 | 7 |
| DriveDiagnosis | 48 | 58,509 | 11 |
| MNIST | 784 | 70,000 | 10 |

**Out-of-distribution detection.** Using the trained generative classifier (i.e., class-conditional distributions obtained from the classifier), the confidence score of each sample can be computed based on the class-conditional probability $P(\mathbf{x}|y = k)$. Taking the log of the probability, we simply define the confidence score $D(\mathbf{x})$ using the Euclidean distance between a test sample and the center of the closest class-conditional distribution in the latent space,

$$D(\mathbf{x}) = -\min_k \|f(\mathbf{x}) - \mathbf{c}_k\|^2. \tag{3}$$

This distance-based confidence score yields discriminative values between ID and OOD samples. In the experiment section, we show that the Euclidean distance in the latent space of our distance classifier is more effective to detect the samples not belonging to the $K$ classes, compared to the Mahalanobis distance in the latent space of the softmax classifier. Moreover, it does not require further computation to obtain the class means and covariance matrix, and the predictive uncertainty can be measured by a single DNN inference.

**Relationship to deep one-class classifier.** Recent studies on one-class classification, which have been mainly applied to anomaly detection, try to employ DNNs in order to effectively model the normality of a single class. Inspired by early work on one-class classification including one-class support vector machine (OC-SVM) (Schölkopf et al., 2001) and support vector data description (SVDD) (Tax & Duin, 2004), Ruff et al. (2018; 2019) proposed a simple yet powerful deep learning objective, DeepSVDD. It trains the DNNs to map samples of the single known class close to its class center in the latent space, showing that it finds a hypersphere of minimum volume with the center $\mathbf{c}$:

$$\min_{\mathcal{W}} \frac{1}{N} \sum_{i=1}^{N} \|f(\mathbf{x}_i; \mathcal{W}) - \mathbf{c}\|^2.$$

Our DNNs with the distance classifier can be interpreted as an extension of DeepSVDD for multi-class classification, which incorporates $K$ one-class classifiers into a single network. In the proposed objective (1), the first term makes the $K$ classifiers distinguishable for the multi-class setting, and the second term learns each classifier by gathering the training samples into their corresponding center, as done in DeepSVDD. The purpose of the one-class classifier is to determine whether a test sample belong to the target class or not, thus training it for each class is useful for detecting out-of-distribution samples in our task as well.

## 3 EXPERIMENTS

In this section, we present experimental results that support the superiority of the proposed model. Using tabular and image datasets, we compare the performance of our distance classifier (i.e., DNNs with dm-layer) with that of the softmax classifier (i.e., DNNs with fc-layer) in terms of both ID classification and OOD detection. We also provide empirical analysis on the effect of our regularization term. Our code and preprocessed datasets will be publicly available for reproducibility.

### 3.1 EVALUATION ON TABULAR DATASETS

**Experimental settings.** We first evaluate our distance classifier using four multi-class tabular datasets with real-valued attributes: GasSensor, Shuttle, DriveDiagnosis, and MNIST. They are downloaded from UCI Machine Learning repository[1], and we use them after preprocessing all the attributes using z-score normalization. Table 1 summarizes the details of the datasets. To simulate

---

[1]https://archive.ics.uci.edu/ml/index.php

Table 2: Performance of ID classification and OOD detection by each confidence score (and the classifier) on tabular datasets. The best results are marked in bold face.

| Data | OOD | Classification acc. | TNR at TPR 85% | AUROC | Detection acc. |
|------|-----|---------------------|----------------|-------|----------------|
|  |  | Baseline (softmax) / Mahalanobis (softmax) / Euclidean (distance) | | | |
| GasSensor | 0 | 99.65 / 99.35 / 99.59 | 42.42 / **95.95** / 90.58 | 57.46 / **95.90** / 94.54 | 64.99 / **91.52** / 88.27 |
| | 1 | 99.71 / 99.33 / 99.64 | 35.00 / 87.82 / **99.20** | 49.54 / 94.86 / **98.99** | 60.96 / 88.65 / **96.03** |
| | 2 | 99.74 / 99.53 / 99.72 | 72.63 / 88.97 / **92.88** | 85.48 / 92.97 / **95.81** | 79.42 / 87.94 / **90.77** |
| | 3 | 99.67 / 99.42 / 99.61 | 93.97 / 33.80 / **99.88** | 95.87 / 75.88 / **99.36** | 90.36 / 72.73 / **97.55** |
| | 4 | 99.72 / 99.54 / 99.73 | 67.31 / 94.99 / **99.49** | 78.67 / 96.65 / **98.91** | 76.62 / 90.94 / **95.88** |
| | 5 | 99.70 / 99.41 / 99.60 | 47.44 / 19.05 / **88.95** | 78.72 / 69.88 / **93.72** | 74.50 / 69.73 / **87.98** |
| Shuttle | 0 | 99.94 / 99.94 / 99.90 | 88.69 / **99.36** / 98.73 | 91.69 / **99.11** / 98.65 | 90.27 / **98.10** / 97.38 |
| | 1 | 99.96 / 99.93 / 99.94 | 69.20 / **100.0** / **100.0** | 77.34 / 99.58 / **99.72** | 79.94 / 99.22 / **97.56** |
| | 2 | 99.96 / 99.96 / 99.96 | 52.63 / 98.83 / **99.53** | 63.63 / 98.51 / **99.00** | 70.58 / 94.75 / **95.39** |
| | 3 | 99.96 / 99.93 / 99.95 | 96.42 / **98.21** / 97.90 | 98.08 / 98.41 / **98.73** | 92.68 / **94.61** / 94.58 |
| | 4 | 99.97 / 99.96 / 99.96 | 69.80 / **100.0** / **100.0** | 76.66 / 99.90 / **99.92** | 80.81 / **99.87** / 99.70 |
| | 5 | 99.95 / 99.93 / 99.94 | 14.00 / **100.0** / **100.0** | 16.84 / **99.94** / 99.93 | 56.01 / **99.92** / 99.91 |
| | 6 | 99.97 / 99.93 / 99.95 | 00.00 / 96.92 / **100.0** | 00.00 / 96.82 / **99.78** | 50.00 / 98.24 / **99.77** |
| DriveDiagnosis | 0 | 99.71 / 98.47 / 99.73 | 13.74 / **82.22** / 62.24 | 20.78 / **91.28** / 80.26 | 51.70 / **84.63** / 74.65 |
| | 1 | 99.72 / 98.89 / 99.84 | 07.98 / 55.94 / **62.22** | 12.09 / 79.74 / **82.97** | 50.04 / 74.44 / **74.92** |
| | 2 | 99.67 / 98.29 / 99.68 | 63.73 / 55.50 / **77.41** | 75.30 / 79.92 / **88.64** | 75.96 / 73.65 / **81.70** |
| | 3 | 99.67 / 98.20 / 99.66 | 53.78 / 80.39 / **89.63** | 63.16 / 90.43 / **94.28** | 69.68 / 84.57 / **88.19** |
| | 4 | 99.74 / 98.59 / 99.73 | 78.71 / 23.42 / **94.07** | 81.72 / 66.72 / **96.55** | 82.44 / 64.03 / **91.21** |
| | 5 | 99.75 / 98.67 / 99.77 | 68.82 / 24.51 / **80.58** | 78.24 / 68.24 / **89.43** | 77.58 / 65.46 / **83.33** |
| | 6 | 99.63 / 98.36 / 99.63 | 08.63 / **99.83** / 91.06 | 10.58 / **99.67** / 95.77 | 50.92 / **98.16** / 90.08 |
| | 7 | 99.68 / 98.31 / 99.75 | 24.62 / 66.13 / **71.97** | 34.04 / **85.34** / 85.68 | 55.47 / **79.10** / 79.30 |
| | 8 | 99.68 / 98.57 / 99.75 | 59.24 / 43.86 / **71.60** | 74.69 / 75.40 / **85.19** | 74.10 / 69.62 / **78.60** |
| | 9 | 99.70 / 98.94 / 99.74 | 02.38 / **65.51** / 28.78 | 04.43 / **84.74** / 63.21 | 50.00 / **77.47** / 61.30 |
| | 10 | 99.61 / 98.23 / 99.61 | 10.06 / **100.0** / **100.0** | 12.93 / **99.97** / 99.99 | 51.89 / **99.97** / 99.87 |
| MNIST | 0 | 97.82 / 96.93 / 97.24 | 85.81 / 63.55 / **89.60** | 82.93 / 84.10 / **93.09** | **87.34** / 76.97 / 87.45 |
| | 1 | 97.86 / 96.87 / 96.99 | **93.34** / 09.77 / 84.20 | **90.39** / 59.52 / 90.75 | **91.47** / 61.69 / 84.93 |
| | 2 | 97.97 / 97.16 / 97.46 | 73.88 / 74.99 / **84.78** | 71.61 / 88.83 / **90.95** | 82.52 / 81.38 / **84.98** |
| | 3 | 97.99 / 97.50 / 97.56 | 72.31 / 49.71 / **89.01** | 69.82 / 79.19 / **92.60** | 81.29 / 72.72 / **87.24** |
| | 4 | 98.02 / 97.26 / 97.47 | 51.20 / 24.91 / **63.24** | 49.36 / 65.96 / **80.15** | 71.27 / 62.61 / **74.74** |
| | 5 | 98.04 / 97.38 / 97.57 | 74.38 / 37.44 / **83.09** | 72.18 / 74.37 / **89.86** | 82.98 / 69.30 / **84.14** |
| | 6 | 97.86 / 96.97 / 97.37 | 73.86 / 49.43 / **87.72** | 71.30 / 79.04 / **91.54** | 81.82 / 72.26 / **85.42** |
| | 7 | 98.10 / 97.17 / 97.51 | 71.20 / 46.75 / **82.05** | 68.90 / 77.26 / **89.60** | 81.10 / 70.32 / **83.59** |
| | 8 | 98.10 / 97.41 / 97.68 | 86.21 / 16.74 / **95.18** | 83.78 / 62.85 / **95.71** | 87.63 / 63.00 / **90.99** |
| | 9 | 98.21 / 97.42 / 97.53 | 78.06 / 12.69 / **87.12** | 75.95 / 57.82 / **91.42** | 84.44 / 60.40 / **86.18** |

the scenario that the test distribution includes both ID and OOD samples, we build the training and test set by regarding one of classes as the OOD class and the rest of them as the ID classes. We exclude the samples of the OOD class from the training set, then train the DNNs using only the ID samples for classifying inputs into the $K$-1 classes. The test set contains all samples of the OOD class as well as the ID samples that are left out for testing. The evaluations are repeated while alternately changing the OOD class, thus we consider $K$ scenarios for each dataset. For all the scenarios, we perform 5-fold cross validation and report the average results.

The multi-layer perceptron (MLP) with three hidden layers is chosen as the DNNs for training the tabular data. For fair comparisons, we employ the same architecture of MLP (# Input attributes $\times 128 \times 128 \times 128 \times$ # Classes) for both the softmax classifier and the distance classifier. We use the Adam optimizer (Kingma & Ba, 2014) with the initial learning rate $\eta = 0.01$, and set the maximum number of epochs to 100. In case of tabular data, we empirically found that the regularization coefficient $\lambda$ hardly affects the performance of our model, so fix it to 1.0 without further hyperparameter tuning.

We consider two competing methods using the DNNs optimized for the softmax classifier: 1) the baseline method (Hendrycks & Gimpel, 2017) uses a maximum value of softmax posterior probability as the confidence score, $\max_k \frac{\exp(\mathbf{w}_k^\top f(\mathbf{x})+b_k)}{\sum_{k'} \exp(\mathbf{w}_{k'}^\top f(\mathbf{x})+b_{k'})}$, and 2) the state-of-the-art method (Lee

Table 3: Performance of OOD detection by each confidence score (and the classifier) on image datasets. The best results are marked in bold face.

| Model | ID | OOD | TNR at TPR 85% | AUROC | Detection acc. |
|---|---|---|---|---|---|
| | | | Baseline (softmax) / Mahalanobis (softmax) / Euclidean (distance) | | |
| ResNet | SVHN | CIFAR-10 | 89.86 / 90.65 / **98.38** | 92.29 / 94.81 / **97.68** | 87.44 / 87.87 / **94.07** |
| | | ImageNet | 91.93 / 85.86 / **98.30** | 93.58 / 92.94 / **97.98** | 88.51 / 85.58 / **93.56** |
| | | LSUN | 90.67 / 85.85 / **95.51** | 92.74 / 92.79 / **95.86** | 87.84 / 85.68 / **90.63** |
| | CIFAR-10 | SVHN | 70.85 / **72.21** / 70.58 | 88.59 / **88.65** / 86.42 | 80.75 / **82.34** / 80.82 |
| | | ImageNet | 83.53 / 67.76 / **98.26** | 91.22 / 86.75 / **96.30** | 85.01 / 79.21 / **91.75** |
| | | LSUN | 89.94 / 73.85 / **98.55** | 93.19 / 88.95 / **97.16** | 87.78 / 82.23 / **92.19** |
| | CIFAR-100 | SVHN | 40.12 / **44.52** / 42.18 | 78.26 / **80.52** / 76.56 | 72.69 / **74.44** / 71.45 |
| | | ImageNet | 44.49 / 48.04 / **50.78** | 78.67 / 76.57 / **82.02** | 72.21 / 69.58 / **75.58** |
| | | LSUN | 44.37 / 46.35 / **51.07** | 78.87 / 76.41 / **83.30** | 72.64 / 69.77 / **77.40** |
| DenseNet | SVHN | CIFAR-10 | 90.06 / 88.65 / **94.26** | 92.98 / 93.67 / **95.49** | 88.05 / 86.85 / **89.67** |
| | | ImageNet | 94.89 / 86.83 / **95.74** | 95.88 / 93.33 / **96.43** | **91.10** / 86.03 / 90.87 |
| | | LSUN | 92.63 / 75.94 / **95.52** | 94.67 / 88.98 / **96.20** | 89.70 / 81.24 / **90.68** |
| | CIFAR-10 | SVHN | 90.63 / 88.32 / **91.31** | 92.87 / 94.06 / **94.85** | 87.52 / 87.09 / **88.33** |
| | | Imagenet | 83.98 / 69.47 / **85.00** | 90.77 / 83.31 / **92.12** | **88.12** / 77.56 / 85.12 |
| | | LSUN | 85.33 / 66.24 / **86.52** | 92.26 / 82.82 / **92.83** | 84.06 / 75.83 / **85.80** |
| | CIFAR-100 | SVHN | 37.80 / 48.96 / **52.48** | 75.14 / 68.82 / **79.16** | 70.03 / 62.02 / **72.37** |
| | | ImageNet | 35.22 / 48.21 / **56.01** | 62.12 / 68.87 / **80.29** | 60.30 / 61.73 / **73.30** |
| | | LSUN | 38.71 / 43.62 / **47.39** | 66.36 / 67.51 / **75.82** | 63.15 / 59.94 / **69.93** |

et al., 2018) defines the score based on the Mahalanobis distance using empirical class means $\hat{\mu}_k$ and covariance matrix $\hat{\Sigma}$, which is $\max_k -(f(\mathbf{x}) - \hat{\mu}_k)^\top \hat{\Sigma}^{-1}(f(\mathbf{x}) - \hat{\mu}_k)$. Note that any OOD samples are not available at training time, so we do not consider advanced calibration techniques for all the methods; for example, temperature scaling, input perturbation (Liang et al., 2018), and regression-based feature ensemble (Lee et al., 2018). We measure the classification accuracy for ID test samples[2], as well as three performance metrics for OOD detection: the true negative rate (TNR) at 85% true positive rate (TPR), the area under the receiver operating characteristic curve (AUROC), and the detection accuracy.[3]

**Experimental results.**  In Table 2, our proposed method (i.e., distance-based confidence score) using the distance classifier considerably outperforms the other competing methods using the softmax classifier in most scenarios. Compared to the baseline method, the Mahalanobis distance-based confidence score sometimes performs better, and sometimes worse. This strongly indicates that the empirical data distribution in the latent space does not always take the form of Gaussian distribution for each class, in case of the softmax classifier. For this reason, our explicit modeling of class-conditional Gaussian distributions using the dm-layer guarantees the GDA assumption, and it eventually helps to distinguish OOD samples from ID samples. Moreover, the distance classifier shows almost the same classification accuracy with the softmax classifier; that is, it improves the performance of OOD detection without compromising the performance of ID classification.

For qualitative comparison on the latent spaces of the softmax classifier and distance classifier, we plot the 2D latent space after training the DNNs whose size of latent dimension is set to 2. Figure 1 illustrates the training and test distributions of the GasSensor dataset, where the class 3 (i.e., Ammonia) is considered as the OOD class. Our DNNs successfully learn the latent space so that ID and OOD samples are separated more clearly than the DNNs of the softmax classifier. Notably, in case of the softmax classifier, the covariance matrices of all the classes are not identical, which violates the necessary condition for the Mahalanobis distance-based confidence score to be effective

---

[2]The state-of-the-art method can predict the class label of test samples by the Mahalanobis distance from class means, $\hat{y}(\mathbf{x}) = \arg\min_k (f(\mathbf{x}) - \hat{\mu}_k)^\top \hat{\Sigma}^{-1}(f(\mathbf{x}) - \hat{\mu}_k)$.

[3]These performance metrics have been mainly used for OOD detection (Lee et al., 2018; Liang et al., 2018).

Table 4: ID classification accuracy of each method on image datasets.

| Model | SVHN | CIFAR-10 | CIFAR-100 |
|---|---|---|---|
| | Baseline (softmax) / Mahalanobis (softmax) / Euclidean (distance) | | |
| ResNet | 95.81 / 95.76 / 95.92 | 93.93 / 93.92 / 94.30 | 75.59 / 74.78 / **78.32** |
| DenseNet | 95.30 / 95.22 / 95.59 | 92.87 / 91.66 / **94.74** | 72.27 / 68.22 / **75.39** |

in detecting OOD samples.[4] In this sense, the proposed score does not require such assumption any longer, because our objective makes the latent space satisfy the GDA assumption.

## 3.2 EVALUATION ON IMAGE DATASETS

**Experimental settings.** We validate the effectiveness of the distance classifier on OOD image detection as well. Two types of deep convolutional neural networks (CNNs) are utilized: ResNet (He et al., 2016) with 100 layers and DenseNet (Huang et al., 2017) with 34 layers. Specifically, we train ResNet and DenseNet for classifying three image datasets: CIFAR-10, CIFAR-100 (Krizhevsky et al., 2009), and SVHN (Netzer et al., 2011). Each dataset used for training the models is considered as ID samples, and the others are considered as OOD samples. To consider a variety of OOD samples at test time, we measure the performance by additionally using TinyImageNet (randomly cropped image patches of size $32 \times 32$ from ImageNet dataset) (Deng et al., 2009) and LSUN (Yu et al., 2015) as test OOD samples. All CNNs are trained with stochastic gradient descent with Nesterov momentum (Duchi et al., 2011), and we follow the training configuration (e.g., the number of epochs, batch size, learning rate and its scheduling, and momentum) suggested by (Lee et al., 2018; Liang et al., 2018). The regularization coefficient $\lambda$ of the distance classifier is set to 0.1.

**Experimental results.** Table 3 shows that our distance classifier also can be generalized well for deeper and more complicated models such as ResNet and DenseNet. Similarly to tabular data, our confidence score achieves the best performance for most test cases, and significantly improves the detection performance over the state-of-the-art method. Interestingly, the distance classifier achieves better ID classification accuracy than the softmax classifier in Table 4. These results show the possibility that any existing DNNs can improve their classification power by adopting the dm-layer, which learns the class centers instead of the class weights. From the experiments, we can conclude that our proposed objective is helpful to accurately classify ID samples as well as identify OOD samples from unknown test distributions.

## 3.3 EFFECT OF REGULARIZATION

We further investigate the effects of our regularization term on the performance and the data distributions in the latent space. We first evaluate the distance classifier, using the DNNs trained with different $\lambda$ values from $10^{-3}$ to $10^3$. Figure 2 presents the performance changes with respect to the $\lambda$ value. In terms of ID classification, the classifier cannot be trained properly when $\lambda$ grows beyond $10^2$, because the regularization term is weighted too much compared to the log posterior term in our objective which learns the decision boundary. On the other hand, we observe that the OOD detection performances are not much affected by the regularization coefficient, unless we set $\lambda$ too small or too large; any values in the range (0.1, 10) are fine enough to obtain the model working well.

We also visualize the 2D latent space where the training distribution of MNIST are represented, varying the value of $\lambda \in \{0.01, 0.1, 1, 10\}$. In Figure 3, even with a small value of $\lambda$, we can find the decision boundary that partitions the space into $K$ regions, whereas the class centers (plotted as black circles) do not match with the actual class means and the samples are spread over the entire space. As $\lambda$ increases, the class centers approach to the actual class means, and simultaneously the samples get closer to its corresponding class center thereby form multiple spheres. As discussed in Section 2, the regularization term enforces the empirical class-conditional distributions to approximate the Gaussian distribution with the mean $\mathbf{c}_k$. In conclusion, the proper value of $\lambda$ makes the DNNs place the class-conditional Gaussian distributions far apart from each other, so the OOD samples are more likely to be located in the rest of the space.

---

[4]This confidence score is derived under the assumption that all the classes share the same covariance matrix.

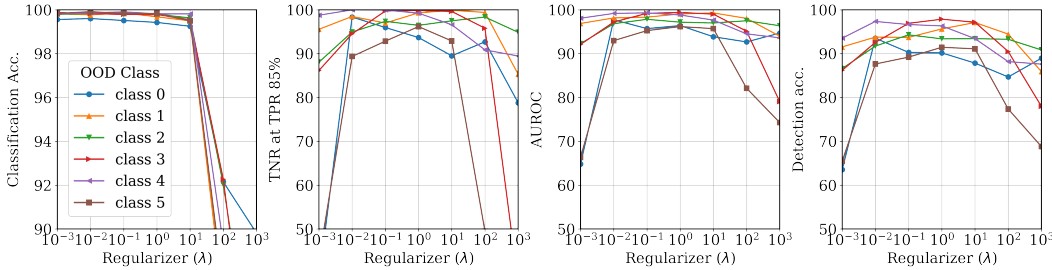

Figure 2: The performance changes with respect to $\lambda$ values. (Dataset: GasSensor)

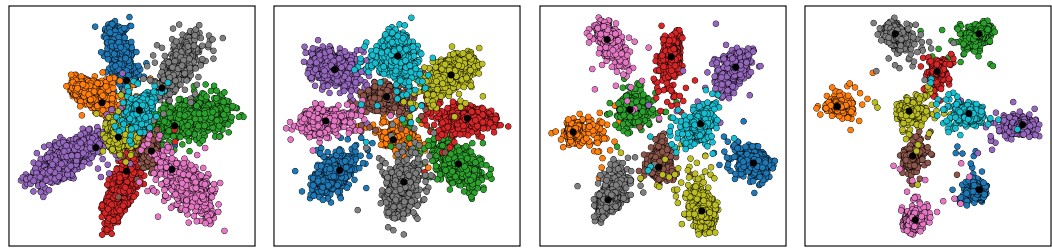

Figure 3: The training distribution in our 2D latent space, $\lambda$=0.01, 0.1, 1, and 10. (Dataset: MNIST)

## 4 RELATED WORK

As DNNs have become the dominant approach to a wide range of real-world applications and the cost of their errors increases rapidly, many studies have been carried out on measuring the uncertainty of a model's prediction, especially for non-Bayesian DNNs (Gal, 2016; Teye et al., 2018). Recently, Malinin & Gales (2018) defined several types of uncertainty, and among them, *distributional uncertainty* occurs by the discrepancy between the training and test distributions. In this sense, the OOD detection task can be understood as modeling the distributional uncertainty, and a variety of approaches have been attempted, including the parameterization of a prior distribution over predictive distributions (Malinin & Gales, 2018) and training multiple classifiers for an ensemble method (Shalev et al., 2018; Vyas et al., 2018).

The baseline method (Hendrycks & Gimpel, 2017) is the first work to define the confidence score by the softmax probability based on a given DNN classifier. To enhance the reliability of detection, ODIN (Liang et al., 2018) applies two calibration techniques, i.e., temperature scaling (Hinton et al., 2015) and input perturbation (Goodfellow et al., 2014), to the baseline method, which can push the softmax scores of ID and OOD samples further apart from each other. Lee et al. (2018) uses the Mahalanobis distance from class means instead of the softmax score, assuming that samples of each class follows the Gaussian distribution in the latent space. However, all of them utilize the DNNs for the discriminative (i.e, softmax) classifier, only optimized for classifying ID samples. Our approach differs from the existing methods in that it explicitly learns the class-conditional Gaussian distributions and computes the score based on the Euclidean distance from class centers.

## 5 CONCLUSION

This paper introduces a deep learning objective to learn the multi-class generative classifier, by fusing the concept of Gaussian discriminant analysis with DNNs. Unlike the conventional softmax classifier, our generative (or distance) classifier learns the class-conditional distributions to be separated from each other and follow the Gaussian distribution at the same time, thus it is able to effectively distinguish OOD samples from ID samples. We empirically show that our confidence score beats other competing methods in detecting both OOD tabular data and OOD images, and also the distance classifier can be easily combined with various types of DNNs to further improve their performances.

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
