# OpenReview forum: "Deep Generative Classifier for Out-of-distribution Sample Detection"
_ICLR.cc/2020/Conference — Reject_

### Official Review · AnonReviewer1 · 2019-10-17
**Official Blind Review #1**

**Rating:** 3

**Review:**

Summary:

Unlike the softmax classifier, the authors considered the generative classifier based on Gaussian discriminative analysis and showed that such deep generative classifiers can be useful for detecting out-of-distribution samples. For various benchmark tasks, the proposed method outperforms baselines based on the softmax classifier.

Detailed comments:

The novelty of this paper is not significant due to the following reasons:

1. The main message (i.e. the concept of the deep generative classifier can be useful for detecting out-of-distribution samples) is not really new because it has been explored before [Lee' 18]. Even though this paper considers training a deep generative classifier directly unlike [Lee' 18], the proposed method looks like a simple variant of [Lee' 18].

2. Missing baselines for training the deep generative classifier: training the deep generative classifier directly has been studied by [Guerriero' 18] and [Pang' 18] but the authors did not compare the proposed training method with such baselines. Because of that, it is hard to say that contributions from proposing a training method are significant.

Questions:

1. Could the authors consider a case without an identity covariance assumption? Most training methods for deep generative classifier assumes the identity covariance matrix because optimizing the log determinant is not easy. So, it would be interesting if the authors can handle this issue.

2. Even though the authors assume the identity covariance matrix, the covariance matrix of pre-trained features can not be an identity matrix. Could the authors report the performance of Mahalanobis detector using the proposed deep generative classifier?

[Lee' 18] Lee, K., Lee, K., Lee, H. and Shin, J., 2018. A simple unified framework for detecting out-of-distribution samples and adversarial attacks. In Advances in Neural Information Processing Systems (pp. 7167-7177).

[Guerriero' 18] Samantha Guerriero, Barbara Caputo, Thomas Mensink, DeepNCM: Deep Nearest Class Mean Classifiers, ICLR workshop 2018.

[Pang' 18] Pang, T., Du, C. and Zhu, J.,  Max-mahalanobis linear discriminant analysis networks. In ICML, 2018.

**Experience Assessment:**

I have published in this field for several years.

**Review Assessment: Checking Correctness Of Derivations And Theory:**

I carefully checked the derivations and theory.

**Review Assessment: Checking Correctness Of Experiments:**

I carefully checked the experiments.

**Review Assessment: Thoroughness In Paper Reading:**

I read the paper thoroughly.

---

> ### Author Response · Authors · 2019-11-07
> **Rebuttal #1**
>
> Thanks for your review.
>
> 1) The main motivation of our work is from the observation that the previous Mahalanobis method adopts the concept of the generative classifier under the strong assumption, which is not realistic enough. Specifically, the latent space optimized by the softmax classifier does not guarantee that each empirical class-conditional distribution follows the Gaussian distribution. In addition, the Mahalanobis detector requires the linear discriminant analysis (LDA) assumption that all the class covariances are the same, to compute the Mahalanobis distance by using tied-covariance matrix. Figure 1 clearly shows that the latent space trained by the softmax classifier is not suitable for the Mahalanobis detector in that 1) all the covariances are not the same as well as 2) ID and OOD samples are difficult to be distinguished.
>
> To address this limitation, we optimize the latent space so that each class-conditional distribution follows the isotropic Gaussian distribution with the same covariance. By doing so, we can simply define the confidence score based on the Euclidean distance without any assumptions, and our proposed score more effectively distinguishes OOD samples from ID samples than the Mahalanobis detector that works on the latent space trained by the softmax classifier.
>
> 2) Thank you for letting us know the missing related work. However, they do not define the confidence score that can be used for detecting OOD samples, so we cannot directly compare their performance. It is worth noting that all of them simply focus on ID classification, not OOD detection (please refer to our responses to the reviewer #3). Unlike the existing distance-based classifiers, our objective introduces the regularization term for OOD detection motivated by [Ruff et al. 2018] and it enables to accurately detect OOD samples.
>
> In the research literature of OOD detection, there have been several attempts to re-train a network based on their own objectives [Malinin and Gales 2018], but they cannot avoid compromising the performance of ID classification. For this reason, the detectors that employ the pre-trained softmax classifier (including the baseline detector and Mahalanobis detector) have gained much attention. In this sense, although our main task is OOD detection, we report the ID classification results in order to emphasize that our classifier succeeds to improve the performance of OOD detection without compromising the ID classification accuracy.
>
>
> About your questions:
>
> 1) Thanks for your suggestion, but it sounds quite challenging to learn the covariance that approximates an arbitrary matrix. By letting it be an identity matrix, our objective can be easily implemented on the deep learning framework as well as efficiently compute the confidence score.
>
> 2) Our proposed objective enforces that the covariance of pre-trained features approximate an identity matrix. Thus, we think the Mahalanobis detector would hardly affect the performance, even though the actual covariance could not be an identity matrix exactly.
>
>
> [Ruff et al. 2018] Deep One-class Classification, ICML 2018
> [Malinin and Gales 2018] Predictive Uncertainty Estimation via Prior Networks, NIPS 2018

---

### Official Review · AnonReviewer3 · 2019-10-26
**Official Blind Review #3**

**Rating:** 3

**Review:**

This paper proposes a metric learning-based generative model for detecting the out-of-distribution examples.  A new objective function is proposed to model class-dependent class-distribution into a Gaussian analysis models. For the proposed objective, the illustration of derived KL divergence under the Gaussian discriminative analysis assumption is well done.  The empirical results conclude the superiority of the proposed loss function in both tabular and image datasets, when comparing the plain network and one with a softmax.

This study aims is to detect out-of-distribution samples for better generalization. However, the related works need to be revised and present the novelty of the work compared to some metric and distance-based learning algorithms. For example, the proposed idea is similar to adding a regularization term to the prototypical network with Euclidean distance (Snell et al. 2016). This aspect is not very well explained.

Another issue is the lack of comparison with state-of-the-art approaches. The Related Work section (Sec. 2) show a baseline (plain) and another one based on softmax. Experimental comparison with state-of-the-art will help to position this work.

** Update ** I read the authors comments and other reviews. Although some clarification were useful, I still maintain my rating of "weak reject", I don't get much excitment and I am not feeling there is something great with this work.


**Experience Assessment:**

I have published one or two papers in this area.

**Review Assessment: Checking Correctness Of Derivations And Theory:**

I assessed the sensibility of the derivations and theory.

**Review Assessment: Checking Correctness Of Experiments:**

I assessed the sensibility of the experiments.

**Review Assessment: Thoroughness In Paper Reading:**

I read the paper at least twice and used my best judgement in assessing the paper.

---

> ### Author Response · Authors · 2019-11-07
> **Rebuttal #3**
>
> Thanks for your review.
>
> 1) We want to emphasize that the most important part of our proposed classifier is the regularization term, because it plays a key role to accurately detect OOD samples. The challenge of the OOD detection task is to obtain the decision boundary between ID samples and OOD samples. To this end, we aim to learn K one-class classifiers that have sphere-shaped decision boundaries with minimum volumes by using the regularization term. DeepSVDD [Ruff et al. 2018] showed that such a sphere-shaped decision boundary is effective to detect abnormal samples in one-class setting, so we extend it to multi-class setting specifically for the OOD detection task. On the other hand, the existing distance-based classifiers including [Snell et al. 2017] only focus on the ID classification based on the distance, so their OOD detection performance would be poor. In Figure 2, the classifier trained with a very small regularization coefficient $\lambda=10^{-3}$ (it seems to be almost the same model with [Snell et al. 2017]) achieves the poor performance in terms of OOD detection while still showing the good performance in terms of ID classification.
>
> 2) In Table 2 and 3, we already compared the performances with the state-of-the-art method, which is Mahalanobis method. [Lee et al. 2018] demonstrated that the Mahalanobis method outperforms both the baseline (plain) and another one (ODIN; equipped with calibration techniques) based on softmax. Furthermore, as we mentioned in the paper, calibration techniques such as temperature scaling and input perturbation are not practical because they require OOD samples from the test distribution to find the best hyperparameter values for OOD detection. For this reason, we omit the comparison with ODIN. Note that the OOD detection performance of our proposed classifier would be much better if any calibration techniques are applied to.
>
> [Ruff et al. 2018] Deep One-class Classification, ICML 2018
> [Snell et al. 2017] Prototypical Networks for Few-shot Learning, NIPS 2017
> [Lee et al. 2018] A Simple Unified Framework for Detecting Out-of-distribution Samples and Adversarial Attacks, NIPS 2018

---

### Official Review · AnonReviewer2 · 2019-10-26
**Official Blind Review #2**

**Rating:** 6

**Review:**

This paper presents an algorithm two learn both classifier and out-of-distribution sample detector. Instead of learning softmax weights, the proposed approach learns to project the inputs to a latent space, where each class is a Gaussian distribution. Out-of-distribution samples can be detected by the distance between the learnt representation and centers. The proposed approach can be viewed as generalization of Gaussian discriminant analysis and one-class classification. The proposed approach is technically sound, and the experiments do show some improvement over previous algorithm on out-of-distribution detection, especially on tabular datasets. However I think there are some weaknesses of this paper

* The novelty is a little thin. The proposed algorithm is based on just a modification of the learning objective, and there are no theoretical analysis of why the proposed approach can work better.
* Experimental result is somewhat weak. Improvement on the image datasets seems marginal, especially on the SVHN dataset. I also doubt if classifying Cifar10 against TinyImageNet or LSUN challenging enough, because these datasets are fairly different. I am interested in whether the proposed approach can detect novel classes, such as training using only 9 of 10 classes on Cifar10.

Another question: does learning classifier as well as centers need additional optimization techniques, like special initialization?

Update
=======

After a careful read of the Mahabolis baseline (Lee et al., 2018) I agree with the authors that this paper has some novelty comparing with previous works, i.e., directly learning a generative classifier instead of converting a discrimitively trained classifier into generative. Combined with the good results obtained. I will raise my score to a weak accept (though without a strong belief).

**Experience Assessment:**

I do not know much about this area.

**Review Assessment: Checking Correctness Of Derivations And Theory:**

I assessed the sensibility of the derivations and theory.

**Review Assessment: Checking Correctness Of Experiments:**

I assessed the sensibility of the experiments.

**Review Assessment: Thoroughness In Paper Reading:**

I read the paper at least twice and used my best judgement in assessing the paper.

---

> ### Author Response · Authors · 2019-11-07
> **Rebuttal #2**
>
> Thanks for your review.
>
> We would disagree with the reviewer on the aspect of novelty. Our work is not about just a modification of the learning objective, but designing a novel objective for effectively detecting OOD samples by using deep neural networks (DNNs) in perspective of Gaussian discriminant analysis (GDA). Unlike the objective used for training the softmax classifier, our proposed objective is theoretically derived from GDA, and this theoretical background guarantees that each class-conditional distribution follows isotropic Gaussian distribution with the same variance in the latent space.
>
> Note that the latent space optimized by the softmax classifier does not guarantee that 1) the class-conditional distributions follow the Gaussian distribution and 2) they have the same covariance matrix, as shown in Figure 1. However, the state-of-the-art method [Lee et al. 2018] computes the Mahalanobis distance using the tied-covariance matrix (assuming that all the covariances are the same) in such space. For this reason, the Mahalanobis method cannot accurately capture the confidence of each sample (i.e., how likely the sample belongs to the in-distribution), and the proposed method clearly address this problem. Thereby, our generative classifier achieves higher OOD detection accuracy than the state-of-the-art method.
>
>
> About your question:
>
> The proposed classifier does not need any additional optimization techniques. In the experiments, we used the Xavier weight initialization and the Adam optimizer, which are conventionally used for DNNs (or the softmax classifier). In this sense, we claim that our learning objective empirically provides the stable convergence and it can be easily employed in the deep learning framework without complicated mathematical modeling or sophisticated optimization.
>
> [Lee et al. 2018] A Simple Unified Framework for Detecting Out-of-distribution Samples and Adversarial Attacks, NIPS 2018

---

### Public Comment · ~Yen-Chang_Hsu1 · 2019-10-01
**A question about how to enforce the covariance to be an identity matrix**

Thanks for the interesting idea! Using a generative classifier for detecting OOD makes lots of sense. It seems that using the GDA assumption while enforcing unit covariance matrix is the key step. Would you elaborate more about how the unit covariance matrix be achieved? The confusion comes from Section 2, in that the loss term derived from KL(P_k||N(c_k, I)) will have an effect of keep minimizing the variance, instead of driving its covariance to be an identity matrix. Is there an empirical observation showing that a unit covariance matrix is achieved by adding this loss term? Thanks!

---

> ### Author Response · Authors · 2019-10-04
> **RE: A question about how to enforce the covariance to be an identity matrix**
>
> Thanks for your comment! The minimization of our KL-divergence term does not guarantee to achieve a unit covariance matrix in the class-conditional distribution, because it is identical to minimizing the variances as you pointed out. We observed that the term \sum ||f(x)-c_k||^2 also can be derived from KL(P_k || N(c_k, \sigma^2 I)) in the same way, i.e., even in the case that we assume the isotropic Gaussian distribution. This phenomenon occurs because of the empirical distribution P_k based on the dirac delta function, which is not continuous but has non-zero values only at the points where the f(x) exists. In this situation, reducing the distance between each point and the class center eventually makes the empirical distribution approximate the isotropic Gaussian distribution, regardless of its assumed variance \sigma. However, it does not affect our overall framework. To use the Euclidean distance for OOD detection and ID classification, the actual values of the variances are not important as long as they are the same for all the classes. In order to control the effect of this KL-divergence term, we introduced the hyperparameter \lambda, which determines the final variance of the class-conditional distributions while interacting with the log posterior term in our objective.

---

### Decision · Program_Chairs · 2019-12-19

**Decision:**

Reject

**Comment:**

The paper presents a training method for deep neural networks to detect out-of-distribution samples under perspective of Gaussian discriminant analysis.

Reviewers and AC agree that some idea is given in the previous work (although it does not focus on training), and additional ideas in the paper are not super novel. Furthermore, experimental results are weak, e.g., comparison with other deep generative classifiers are desirable, as the paper focuses on training such deep models.

Hence, I recommend rejection.